

**Contrasting responses of phytoplankton productivity between coastal and offshore**
**surface waters in the Taiwan Strait and the South China Sea to future CO₂-induced**
**acidification**
Guang Gao[1], Tifeng Wang[1], Jiazhen Sun[1], Xin Zhao[1], Lifang Wang[1], Xianghui Guo[1],
Kunshan Gao[1,2]*
[1]State Key Laboratory of Marine Environmental Science & College of Ocean and Earth
Sciences, Xiamen University, Xiamen 361005, China
[2]Co-Innovation Center of Jiangsu Marine Bio-industry Technology, Jiangsu Ocean
University, Lianyungang 222005, China
*Corresponding author: ksgao@xmu.edu.cn



**Abstract**
Future $CO_2$-induced ocean acidification (OA) has been documented to either inhibit or
enhance or result in no effect on marine primary productivity (PP). In order to examine
effects of OA under multiple drivers, we investigated the influences of OA (a decrease of
0.4 $pH_{total}$ units with corresponding $CO_2$ concentrations ranged 22.0-39.7 μM) on PP
through deck-incubation experiments at 101 stations in the Taiwan Strait and the South
China Sea (SCS), including the coastal zone, the continental shelf and slope, as well as
deep-water basin. The daily net primary productivities in surface seawater under incident
solar radiation ranged from 17-306 μg C (μg Chl $a$)$^{-1}$ d$^{-1}$, with the responses of PP to OA
being region-dependent and the OA-induced changes varying from -88.03% (inhibition)
to 56.87% (enhancement). The OA-treatment stimulated PP in surface waters of coastal,
estuarine and shelf waters, but suppressed it in the South China Sea basin. Such
OA-induced changes in PP were significantly related to $NO_X$ (the sum of $NO_3^-$ and $NO_2^-$)
availability, in situ pH and solar radiation in surface seawater, but negatively related to
salinity changes. Our results indicate that phytoplankton cells are more vulnerable to pH
drop in oligotrophic waters. Considering high nutrient and low salinity in coastal waters
and reduced nutrient availability in pelagic zones with the progressive stratification
associated with ocean warming, our results imply that future OA will enhance PP in
coastal waters but decrease it in pelagic oligotrophic zones.
**Keywords:** $CO_2$; Taiwan Strait; ocean acidification; photosynthesis; primary productivity;



South China Sea
**1 Introduction**

The oceans have absorbed about one-third of anthropogenically released $CO_2$, which

increased dissolved $CO_2$ and decreased pH of seawater (Gattuso et al., 2015), leading to
ocean acidification (OA). OA has been shown to result in profound influences on marine
ecosystems (see the reviews and literature therein, Mostofa et al., 2016; Doney et al.,
2020). Marine photosynthetic organisms, which contribute about half of the global
primary production, are also being affected by OA (see the reviews and literatures therein,
Riebesell et al., 2018; Gao et al., 2019a). It is of general concern that the oceans are going
to take more or less $CO_2$ with progressive OA, since the amount of $CO_2$ uptake by the
oceans is essential to predict global and ocean warming trends. Therefore, it is important
to understand the responses of the key players of marine biological $CO_2$ pump, the
phytoplankton, to OA and other climate change drivers.

Elevated $CO_2$ is well recognized to lessen the dependence of algae and

cyanobacteria on energy-consuming $CO_2$ concentrating mechanisms (CCMs) which
concentrate $CO_2$ around Rubisco, the key site for photosynthetic carbon fixation (Raven
& Beardall, 2014 and references therein; Hennon et al., 2015). The energy freed up from
the down-regulated CCMs under increased $CO_2$ concentrations can be applied to other
metabolic processes, resulting in a modest increase in algal growth (Wu et al., 2010;
Hopkinson et al., 2011; Xu et al., 2017). Accordingly, elevated $CO_2$ availability could





potentially enhance marine primary productivity (Schippers et al., 2004). For instance,
across 18 stations in the central Atlantic Ocean primary productivity was stimulated by
15-19% under elevated dissolved $CO_2$ concentrations up to 36 μM (Hein and
Sand-Jensen 1997). On the other hand, neutral effects of OA on growth rates of
phytoplankton communities were reported in five of six $CO_2$ manipulation experiments in
the coastal Pacific (Tortell et al., 2000). Furthermore, simulated future OA reduced
surface PP in pelagic surface waters of Northern SCS and East China Sea (Gao et al.,
2012). It seems that the impacts of OA on PP could be region-dependent. The varying
effects of OA may be related to the regulation of other factors such as light intensity (Gao
et al., 2012), temperature (Holding et al., 2015), nutrients (Tremblay et al., 2006) and
community structure (Dutkiewicz et al., 2015).

Taiwan Strait of the East China Sea, located between southeast Mainland China and

the Taiwan Island, is an important channel in transporting water and biogenic elements
between the East China Sea (ECS) and the South China Sea (SCS). Among the Chinese
coastal areas, the Taiwan Strait is distinguished by its unique location. In addition to
riverine inputs, it also receives nutrients from upwelling (Hong et al., 2011). Primary
productivity is much higher in coastal waters than that in pelagic zones due to increased
supply of nutrients through river runoff and upwelling (Chen, 2003; Cloern et al., 2014).
The South China Sea (SCS), located from the equator to 23.8 °N, from 99.1 to 121.1 °E
and encompassing an area of about $3.5 \times 10^6 \, km^2$, is one of the largest marginal seas in





the world. As a marginal sea of the Western Pacific Ocean, it has a deep semi-closed
basin (with depths > 5000 m) and wide continental shelves, characterized by a tropical
and subtropical climate (Jin et al., 2016). Approximately 80% of ocean organic carbon is
buried in the Earth's continental shelves and therefore continental margins play an
essential role in the ocean carbon cycle (Hedges & Keil, 1995). Investigating how ocean
acidification affects primary productivity in the Taiwan Strait and the SCS could help us
to understand the contribution of marginal seas to carbon sink under the future
$CO_2$-increased scenarios. Although small-scale studies on OA impacts have been
conducted in the ESC and the SCS (Gao et al., 2012, 2017), our understanding of how
OA affects PP in marginal seas is still fragmentary and superficial. In this study, we
conducted three cruises in the Taiwan Strait and the SCS, covering an area of $8.3 \times 10^5$
$km^2$, and aimed to provide in-depth insight into how OA and/or episodic $pCO_2$ rise
affects PP in marginal seas with comparisons to other types of waters.
**2 Materials and Methods**
**2.1 Investigation areas**
To study the impacts of projected OA (dropping by ~0.4 pH) on marine primary
productivity in different areas, we carried out deck-based experiments during the 3
cruises supported by National Natural Science Foundation of China (NSFC), which took
place in the Taiwan Strait (Jul 14th-25th, 2016), the South China Sea basin (Sep 6-24th,
2016), and the West South China Sea (Sep 14th to Oct 24th, 2017), respectively. The





experiments were conducted at 101 stations with coverage of 12 $^{o}$N-26 $^{o}$N and 110
$^{o}$E-120 $^{o}$E (Fig. 1). Investigation areas include the coastal zone (< 50 m), the continental
shelf (50-200 m) and the slope (200-1000 m), and the vast deep-water basin (> 1000 m).
**2.2 Measurements of temperature and carbonate chemistry parameters**
The temperature and salinity of surface seawater at each station were monitored with
an onboard CTD (Seabird, USA). $pH_{NBS}$ was measured with an Orion 2-Star pH meter
(Thermo scientific, USA) that was calibrated with standard National Bureau of Standards
(NBS) buffers (pH=4.01, 7.00, and 10.01 at 25.0 $^{o}$C; Thermo Fisher Scientific Inc., USA).
The analytical precision was ±0.001. Total alkalinity (TAlK) was determined using Gran
titration on a 25-mL sample with a TA analyzer (AS-ALK1, Apollo SciTech, USA) that
was regularly calibrated with certified reference materials supplied by A. G. Dickson at
the Scripps Institution of Oceanography (Gao et al., 2018a). The analytical precision was
$\pm 2$ µmol kg$^{-1}$. $CO_2$ concentration in seawater and the $pH_{Total}$ ($pH_T$) values was calculated
by using CO2SYS (Pierrot et al., 2006) with the input of $pH_{NBS}$ and TAlK data.
**2.3 Nutrient measurement**
Surface seawater was collected from the Conductivity Temperature Depth (CTD)
rosette/Niskin bottles with a clean 125 mL HDPE (High-Density Polyethylene) sample
container. The nitrate and nitrite concentrations in seawater were then measured with a
Technicon AA3 Auto-Analyzer (Bran-Lube, GmbH, Germany). The quantitative limits
for nitrate and nitrite were 0.1 µmol L$^{-1}$ and 0.04 µmol L$^{-1}$, respectively. We used





certified reference materials (CRMS) (https://www.jamstec.go.jp/scor/) as external
quality checks, and the analytical precision was better than ±1% during the whole cruise.
Nutrient measurement was conducted in the cruise of the South China Sea basin. Due to
the limit of human resources, it was not conducted in the other two cruises.
**2.4 Solar radiation**

The incident solar radiation intensity during the cruises was recorded with an

Eldonet broadband filter radiometer (Eldonet XP, Real Time Computer, Germany). This
device has three channels for PAR (400–700 nm), UV-A (315–400 nm) and UV-B (280–
315 nm) irradiance, respectively, which records the means of solar radiations over each
minute. The instrument was fixed at the top layer of the ship to avoid shading.
**2.5 Determination of primary productivity**

Surface seawater (0-1m) was collected a 10 L acid-cleaned (1 M HCl) plastic bucket

and pre-filtered (200 μm mesh size) to remove large grazers. To prepare high $CO_2$ (HC)
seawater, $CO_2$-saturated seawater was added into pre-filtered seawater until a decrease of
~0.4 units in pH (corresponding $CO_2$ concentrations being 22.0-39.7 μM) was
approached (Gattuso et al., 2010). The same amount of filtered seawater (0.22 μm) was
added into the pre-filtered seawater setting as ambient $CO_2$ (AC) control. Prepared AC
and HC seawater was allocated into 50-mL quartz tubes in triplicate, inoculated with 5
μCi (0.185 MBq) $NaH^{14}CO_3$ (ICN Radiochemicals, USA), and then incubated for 24 h
(over a day-night cycle) under 100 % incident solar irradiances in a water bath for





temperature control by running through surface seawater. After the incubation, the cells
were filtered onto GF/F filters (Whatman) and immediately frozen at −20 °C for later
analysis. In the laboratory, the frozen filters were transferred to 20 mL scintillation vials,
thawed and exposed to HCl fumes for 12 h, and dried (55 ℃, 6 h) to expel non-fixed $^{14}$C,
as previously reported (Gao et al., 2017). Then 3 mL scintillation cocktail (Perkin
Elmer®, OptiPhase HiSafe) was added to each vial. After 2 h of reaction, the
incorporated radioactivity was counted by a liquid scintillation counting (LS 6500,
Beckman Coulter, USA). The carbon fixation for 24 h incubation was taken as
chlorophyll (Chl) $a$-normalized daily net primary productivity (PP, µg C (µg Chl $a$)$^{-1}$)
(Gao et al., 2017). The changes (%) of PP induced by ocean acidification were expressed
as (PP$_{HC}$-PP$_{AC}$)/PP$_{AC}$×100, where PP$_{HC}$ and PP$_{AC}$ are the net daily primary productivity
under HC and AC, respectively.
**2.6 Chl $a$ measurement**

Pre-filtered (200 µm mesh size) surface seawater (500-2000 mL) at each station was

filtered onto GF/F filter (25 mm, Whatman) and then stored at -80 $^{o}$C. After returning to
laboratory, phytoplankton cells on the GF/F filter were extracted overnight in absolute
methanol at 4 $^{o}$C in darkness. After centrifugation (5000 $g$ for 10 min), the absorption
values of the supernatants were analyzed by a UV–VIS spectrophotometer (DU800,
Beckman, Fullerton, California, USA). The concentration of chlorophylls $a$ (Chl $a$) was
calculated according to Porra (2002).





**2.7 Data analysis**

The data of environmental parameters were expressed in raw and the data of PP were

the means of triplicate incubations. Two-way analysis of variance (ANOVA) was used to
analyze the effects of OA and location on PP. Least significant difference (LSD) was used
to for *post hoc* analysis. Linear fitting analysis was conducted with Pearson correlation
analysis to assess the relationship between PP and environmental factors. A 95%
confidence level was used in all analyses.
**3 Results**

During the cruises, surface temperature ranged from 25.0 to 29.9 $^{\circ}$C in the Taiwan

Strait and from 27.1 to 30.2 $^{\circ}$C in the South China Sea (Fig. 2a). Surface salinity ranged
from 30.0 to 34.0 in the Taiwan Strait and from 31.0 to 34.3 in the South China Sea (Fig.
2b). The lower salinities were found in the estuaries of Minjiang and Jiulong Rivers as
well as Mekong River-induced Rip current. High salinities were found in the SCS basin.
Surface $pH_T$ changed between 7.91-8.20 in the Taiwan Strait with the higher values in the
estuary of Minjiang River (Fig. 2c). On the contrary, surface pH had a narrower range
(8.06-8.23) in the South China Sea and the lower values occurred near the islands in the
Philippines. TAlK ranged from 2100 to 2359 $\mu mol\ L^{-1}$ in the Taiwan Strait and 2126 to
2369 $\mu mol\ L^{-1}$ in the South China Sea (Fig. 2d). The lowest value occurred in the estuary
of Minjiang River. $CO_2$ concentration in surface seawater changed from 6.4-15.9 $\mu M\ kg^{-1}$
SW in the Taiwan Strait, and 9.3-14.3 $\mu M\ kg^{-1}$ SW in the SCS (Fig. 1e). It showed an



opposite pattern to surface pH, with the lowest value in the estuary of Minjiang River in
the Taiwan Strait and highest value in near the islands in the Philippines in the South
China Sea. During the PP investigation period, the daytime mean PAR intensity ranged
from 126.6 to 145.2 W m$^{-2}$ s$^{-1}$ in the Taiwan Strait and 37.3 to 150.0 W m$^{-2}$ s$^{-1}$ in the SCS
(Fig. 2f).

The concentration of Chl *a* ranged from 0.11 to 12.13 μg L$^{-1}$ in the Taiwan Strait (Fig.

3). The highest concentration occurred in the estuary of the Minjiang River. The
concentration of Chl *a* in the SCS ranged from 0.037 to 7.43 μg L$^{-1}$. The highest
concentration was found in the coastal areas of Guangdong province in China. For both
the Taiwan Strait and the SCS, there were high Chl *a* concentrations (> 1.0 μg L$^{-1}$) in
coastal areas, particularly in the estuaries of the Minjing River, Jiulong River and Pearl
River. On the contrary, Chl *a* concentrations in offshore areas were lower than 0.2 μg L$^{-1}$.

Surface primary productivity changed from 99-302 μg C (μg Chl *a*)$^{-1}$ d$^{-1}$ in the

Taiwan Strait, and from 17-306 μg C (μg Chl *a*)$^{-1}$ d$^{-1}$ in the South China Sea (Fig. 4).
High surface primary productivity (> 200 μg C (μg Chl *a*)$^{-1}$ d$^{-1}$) was found in the
estuaries of the Minjing River, Jiulong River, and Pearl River and areas near the East of
Vietnam. In pelagic zones, the surface primary productivity was usually lower than 100
μg C (μg Chl *a*)$^{-1}$ d$^{-1}$.

Through a series of onboard $CO_2$-enrich experiments we observed that effects of the

elevated p$CO_2$ on primary productivity of surface phytoplankton community ranged from



-88.03% (inhibition) to 56.87% (promotion), revealing significant regional differences
(ANOVA, $F_{(100, 404)} = 4.103$, $p < 0.001$, Fig. 5). Among 101 stations, 70 stations showed
insignificant OA effects. OA increased PP at 6 stations and reduced PP at 25 stations.
Positive effects of OA on surface primary productivity was observed in the Taiwan Strait
and the western SCS (Fig. 5, red-yellow shading areas), with the maximal enhancement
of 56.9% in the station approaching Mekong River plume (LSD, $p < 0.001$). Reduction in
PP induced by the elevated $CO_2$ was mainly found in the central SCS basin within the
latitudes of 10 $^o$N to14 $^o$N and the longitudes of 114.5 $^o$E to 118 $^o$E (Fig. 5, blue-purple
shading areas), with inhibition rates ranging from 24.02% to 88.03% (Fig. 5, LSD, $p <$
0.05). These results showed a region-related effect of OA on photosynthetic carbon
fixation of surface phytoplankton assemblages. Overall, the elevated $pCO_2$ had neutral or
positive effects on primary productivity in nearshore waters, while having adverse effects
in pelagic waters.

By analyzing the correlations between OA-induced PP changes and regional

environmental parameters, we found that OA-induced changes in phytoplankton primary
productivity was significantly positively related with *in situ* pH ($p < 0.001$, $r = 0.379$),
NOx availability (the concentrations of $NO_3^-$ + $NO_2^-$ at the bottom of upper mixing layers
as they were unmeasurable in the surface water, $p = 0.002$, $r = 0.727$), PAR density ($p =$
0.002, $r = 0.311$) and primary productivity ($p = 0.004$, $r = 0.284$) (Fig. 6 and Table S1).
On the other hand, the influence induced by OA was negatively related to salinity that



ranged from 30.00 to 34.28 ($p < 0.001$, $r = -0.418$).

**4 Discussion**

In the present study, we found that the elevated $pCO_2$ and associated pH drop

increased or did not affect PP in coastal waters but reduced it in pelagic waters. Our
results suggested that the enhanced effects of the OA treatment on photosynthetic carbon
fixation depend on regions of different physicochemical conditions. Higher levels of
nutrients due to runoffs or upwellings should be mainly responsible for the enhancement.
On the other hand, such stimulation could be related to higher UV-attenuation in these
coastal waters that contain more organic matters (Hader et al., 2015), since we employed
UV-transparent vessels for the incubations. In addition, coastal diatoms appear to benefit
more from OA than pelagic ones (Li et al., 2016). Therefore, community structure
differences might also be responsible for the differences of the short-term high
$CO_2$-induced acidification between coastal and pelagic waters.

OA is deemed to have two kinds of effects at least (Xu et al., 2017; Shi et al., 2019).

The first one is the enrichment of $CO_2$, which is usually beneficial for photosynthetic
carbon fixation and growth of algae because insufficient ambient $CO_2$ limits algal
photosynthesis (Hein & Sand-Jensen, 1997; Bach & Taucher, 2019). The other effect is
the decreased pH which could be harmful because it disturbs the acid-base balance
between extracellular and intracellular environments. For instance, the decreased pH
projected for future OA was shown to reduce the growth of the diazotroph



*Trichodesmium* (Hong et al., 2017), decrease PSII activity by reducing removal rate of
PsbD (D2) (Gao et al., 2018b) and increase mitochondrial and photo-respirations in
diatoms and phytoplankton assemblages (Yang and Gao 2012, Jin et al., 2015). In
addition, OA could reduce the RuBisCO transcription of diatoms, which also contributed
to the decreased growth (Endo et al., 2015). Therefore, the net impact of OA depends on
the balance between its positive and negative effects, leading to enhanced, inhibited or
neutral influences, as reported in diatoms (Gao et al., 2012, Li et al., 2021) and
phytoplankton assemblages in the Arctic and subarctic shelf seas (Hoppe et al., 2018), the
North Sea (Eberlein et al., 2017) and the South China Sea (Wu and Gao 2010, Gao et al.,

2012).

In the present study, OA increased or did not affect PP in coastal waters but reduced

it in offshore waters. This is significantly related to nutrient availability (Fig. 6d), with
that the inhibitory effect was minimized when NOx availability increased. Riverine
inputs, including the Minjiang River, Jiulong River, Pearl River, and Mekong River, are
the primary source of nutrients in the coastal and shelf zones, resulting in higher
concentrations of nutrients and lower salinity in these waters (Xiao et al., 2018). It was
reported that elevated $pCO_2$ decreased net organic carbon production of
natural plankton community in nutrient-depleted waters (Yoshimura et al., 2010).
Furthermore, OA did not affect the specific growth rate of a diatom under N-replete
condition but reduced it under N-limited condition (Li et al., 2018). The alleviating effect





of nutrient enrichment on OA-induced stress could be multifaceted. Firstly, algae could
cope with the acid-base perturbation caused by OA through active proton pumps
(McNicholl et al., 2019). The operation of such proton pumps need some essential
proteins, such as plasma membrane H$^+$-ATPase, whose synthesis is nutrient-dependent
(Taylor et al., 2012; Xu et al., 2017). Secondly, it has been shown that nutrient
enrichment could accelerate the repair rate of PSII via synthesizing the key proteins such
as PsbA (D1), and PsbD (D2) (Geider et al., 1993; Li et al., 2015). Thirdly, nitrogen
enrichment could significantly increase the synthesis and content of photosynthetic
pigments including Chl *a*, phycocyanin, and phycoerythrin (Johnson & Carpenter, 2018;
Gao et al., 2019b), contributing to high photosynthetic activity under stressful
environmental conditions. Negative correlation between OA-induced changes of PP and
salinity was found in this study. While little has been documented on the relationship
between salinity and OA (Wulff et al., 2018; Sugie et al., 2020; Xu et al., 2020), lowered
salinity has been shown to alleviate the impact of OA on a coccolithorphorid (Xu et al.,
2020). Nevertheless, we presume the enhanced PP could mainly be related to nutrient
availability because lower salinity in coastal waters usually companies with high nutrient
levels (Li et al., 2011). In addition, local pH may be another factor that affects the
impacts of OA. There are diurnal and seasonal fluctuations of pH in coastal waters and
phytoplankton that adapt well to the fluctuant pH environments would be tolerant to the
decreased pH caused by OA (Flynn et al., 2012, Li et al., 2016). On the other hand, the



surface pH in the ocean basin is relatively stable, with a varied range of only ~0.024 over
a month (Hofmann et al., 2011). Therefore, the phytoplankton cells living in these
environments could be more sensitive to pH drop due to elevated $pCO_2$ (Li et al., 2016).

The specific environmental conditions have profound effects on shaping diverse

dominant phytoplankton groups (Boyd et al., 2010). Larger eukaryotic groups (especially
diatoms) usually dominate the complex coastal regions, while picophytoplankton
(*Prochlorococcus* and *Synechococcus*), characterizing with more efficient nutrients
uptake, dominate the relatively stable offshore waters (Dutkiewicz et al., 2015). In
summer and early autumn, previous investigations demonstrated that diatoms dominated
in the northern waters and the Taiwan Strait (coastal and shelf regions) with the high
abundance of phytoplankton, which are consistent with our Chl *a* data; *Prochlorococcus*
and *Synechococcus* dominated in the SCS basin and the north of SCS (slope and basin
regions) (Xiao et al., 2018, Zhong et al., 2020). In addition, it has been reported that
larger cells benefit more from OA because a thicker diffusion layer around the cells limits
the transport of $CO_2$ (Feng et al., 2010; Wu et al., 2014). In contrast, a thinner diffusion
layer and higher surface to volume ratio in smaller phytoplankton cells can make them
easier to transport $CO_2$ near the cell surface and within the cells, and therefore
picophytoplankton species are less $CO_2$-limited (Bao and Gao, 2021). Therefore, different
community structures between coastal and pelagic areas could also be responsible for the
enhanced and inhibitory effects of OA.





*Conclusions*

By investigating the impacts of the elevated $pCO_2$ on PP in the Taiwan Strait and the

SCS, we demonstrated that such short OA-treatments induced changes in PP were mainly

related to NOx availability based on Pearson correlation coefficients, supporting the

hypothesis that negative impacts of OA on PP increase from coastal to pelagic waters

(Gao et al., 2019a). In view of ocean climate changes, strengthened stratification due to

global warming would reduce the upward transports of nutrients and further reduce

nutrient availability, consequently, leading to exacerbating impacts of OA on PP in

pelagic zones. Meanwhile, PP in coastal and/or upwelled waters would be stimulated or

non-affected by OA with continuous discharges of nutrients from terrestrial environments,

which may imply higher PP and enhance frequency of harmful algal blooms in future

oceans.

*Data availability.* All data are included in the article or Supplement.

*Author contributions.* KG and TW developed the original idea and designed research.

TW and JS carried out fieldwork. GG provided statistical analyses and prepared figures.

GG, KG, and XZ wrote the manuscript. All contributed to revising the paper.

*Competing interests.* The contact author has declared that neither they nor their

co-authors have any competing interests.

*Disclaimer.* Publisher's note: Copernicus Publications remains neutral with regard to

jurisdictional claims in published maps and institutional affiliations.



*Acknowledgements.*This work was supported by the National Natural Science Foundation
of China (41720104005, 41890803 and 42076154), and the Fundamental Research Funds
for the Central Universities (20720200111). We appreciate the NFSC Shiptime Sharing
Project (project number: 41849901) for supporting the Taiwan Strait cruise
(NORC2016-04). We appreciate the chief scientists Yihua Cai, Huabin Mao and Chen Shi
and the R/V Yanping II, Shiyan I and Shiyan III for leading and conducting the cruises.

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



**Figure captions**

**Fig. 1** Sampling stations for the incubation experiments in the Taiwan Strait and the South China Sea during three cruises. Taiwan Strait cruise was conducted in July 2016 (red dots), South China Sea Basin cruise were conducted in September 2016 (blue dots) and Western South China Sea cruise was conducted in September 2017 (black dots). The arrows represent surface circulation fields in summer in the vicinity of Vietnam coast based on Lan et al. (2006).

**Fig. 2** Temperature ($^{o}$C, panel a), salinity (panel b), pH (panel c), total alkalinity (μmol L$^{-1}$, panel d), and $CO_2$ (μmol kg$^{-1}$ SW, panel e) in surface seawater and mean PAR intensity (W m$^{-2}$ s$^{-1}$, panel f) during the PP incubation experiments.

**Fig. 3** Chl $a$ concentration (μg L$^{-1}$) in the Taiwan Strait and the South China Sea during research cruises.

**Fig. 4** Surface primary productivity (μg C (μg Chl $a$)$^{-1}$ d$^{-1}$) in the Taiwan Strait and the South China Sea during research cruises.

**Fig. 5** Ocean acidification (pH decreases of 0.4 units) induced changes (%) of surface primary productivity in the Taiwan Strait and the South China Sea. Red-yellow shading represents a positive effect on PP and blue-purple shading represents a negative effect. Positive effect was found in coastal waters and estuary affected waters, such as the Taiwan Strait, the Pearl River plume, Mekong River induced Rip current in West China Sea. Negative effect was found in surface of oligotrophic waters like SCS Basin.





**Fig. 6** Ocean acidification (pH decreases of 0.4 units) induced changes (%) on surface

primary productivity in the South China Sea as a function of salinity (a), PAR (b),

ambient pH (c), and nitrate plus nitrite concentration (d). The dotted lines represent

95% confidence intervals.



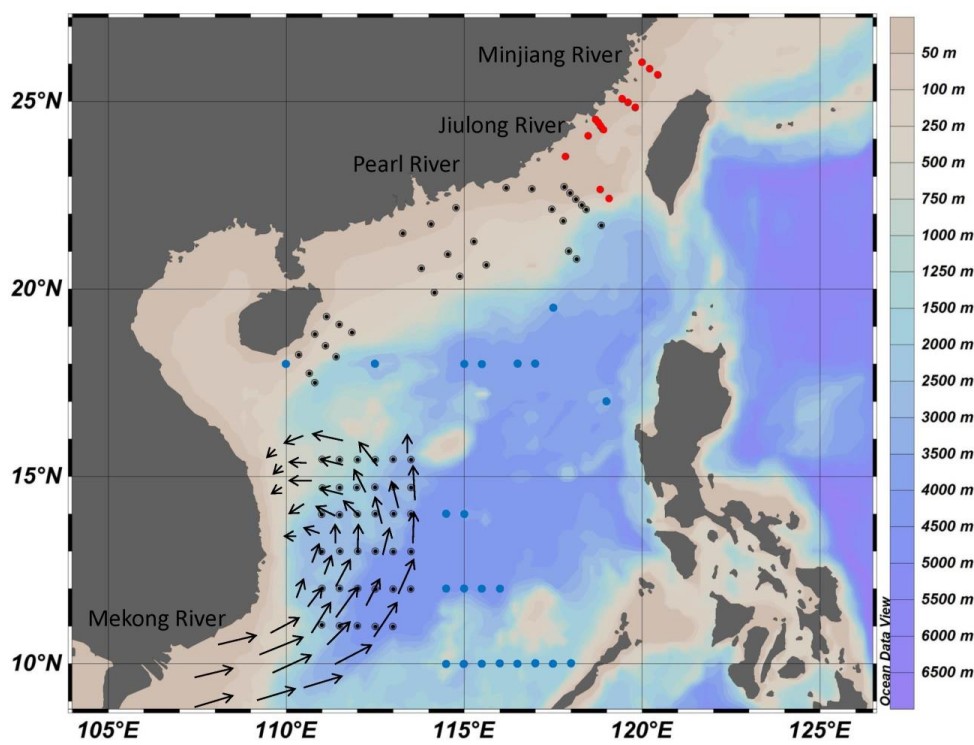

Fig. 1





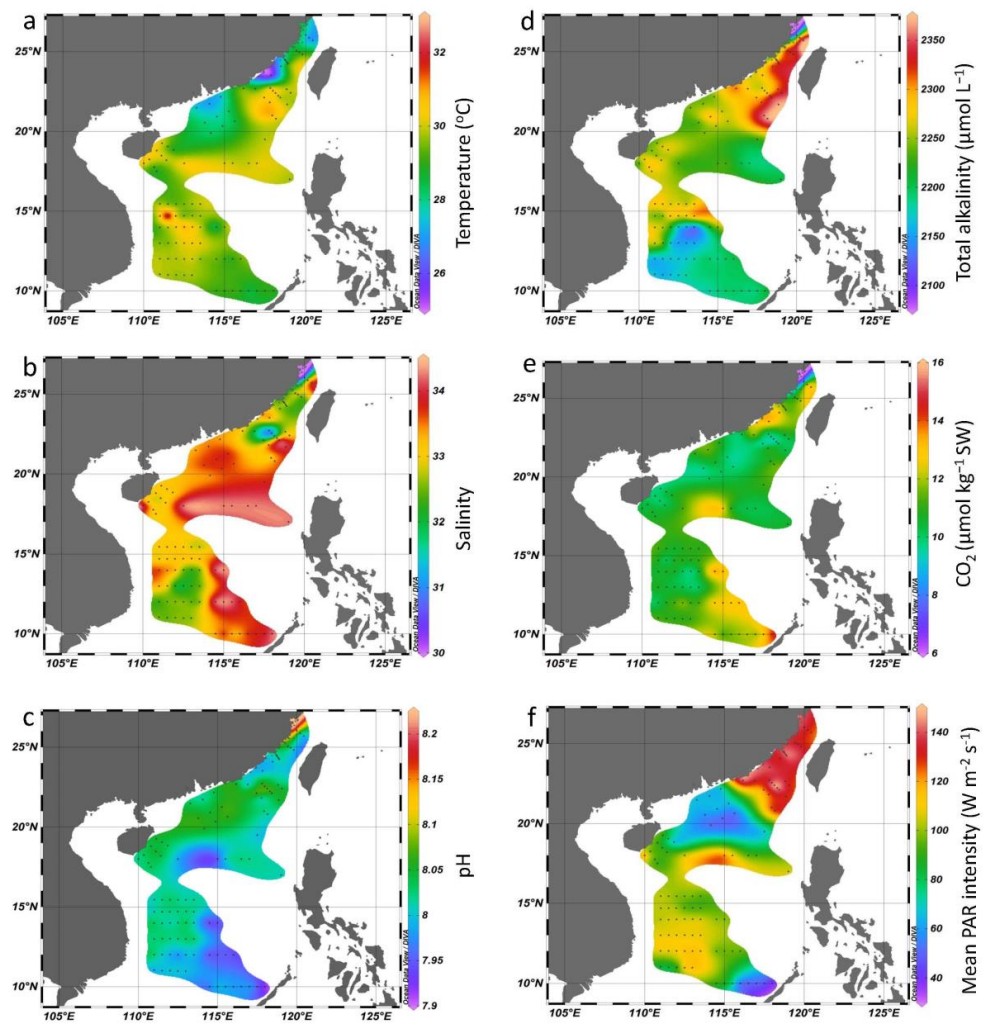

Fig. 2





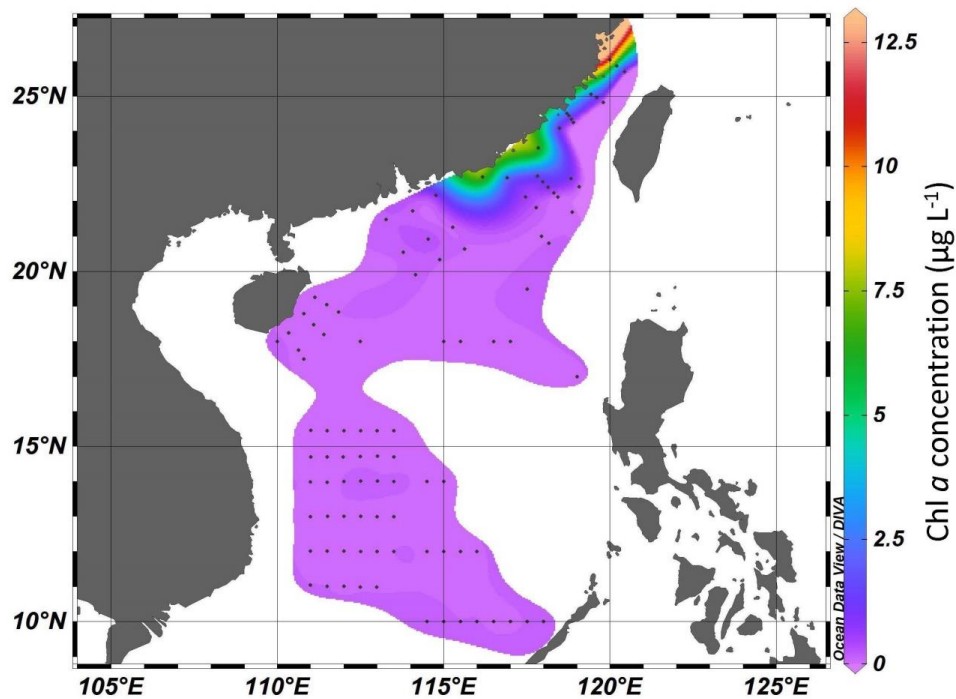

Fig. 3



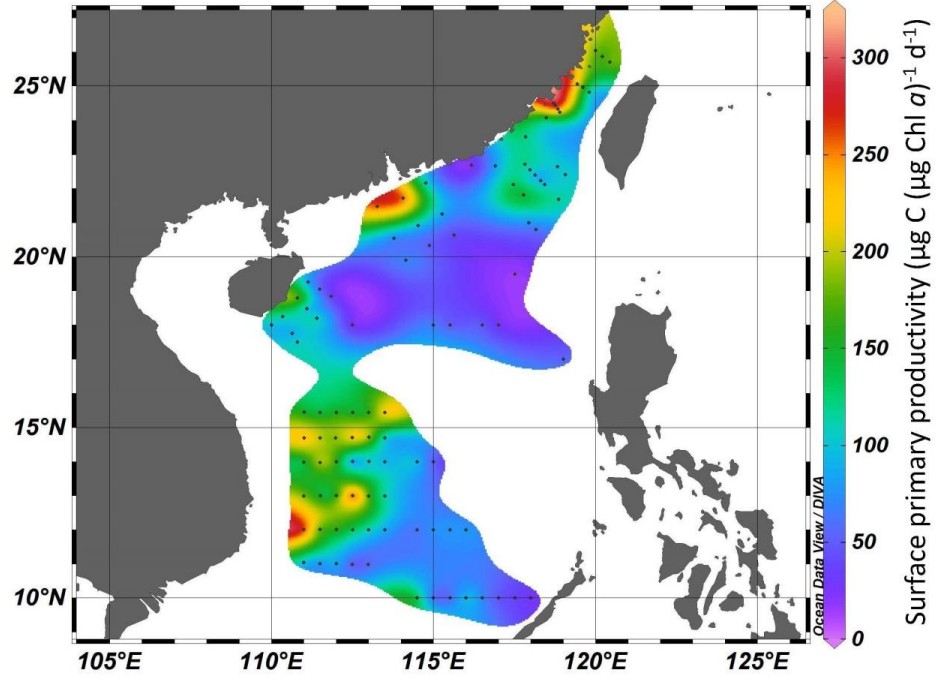

Fig. 4



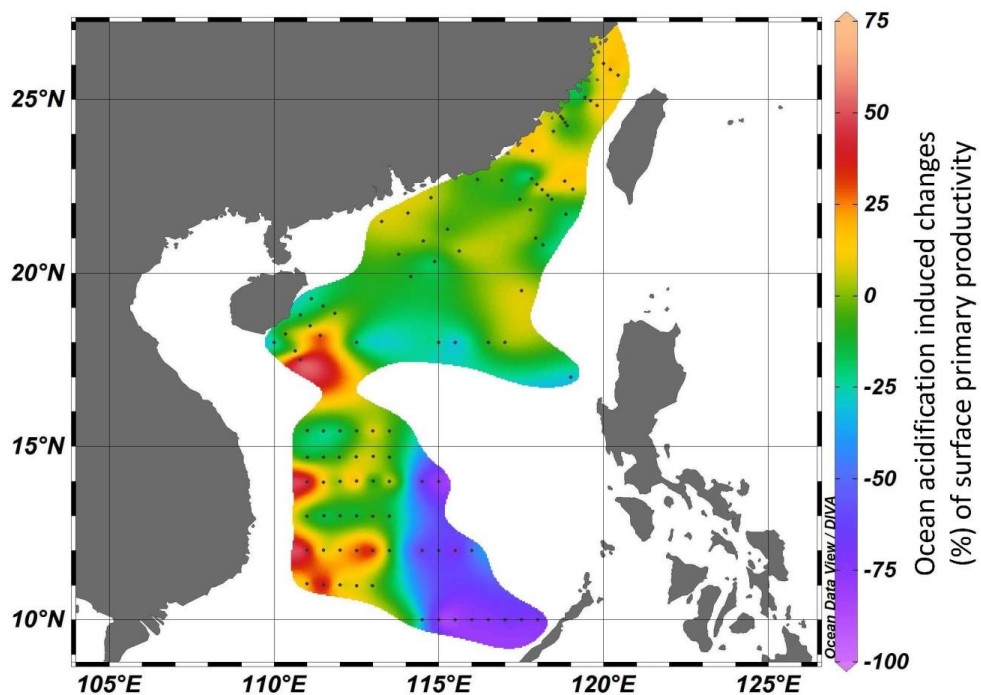

Fig. 5





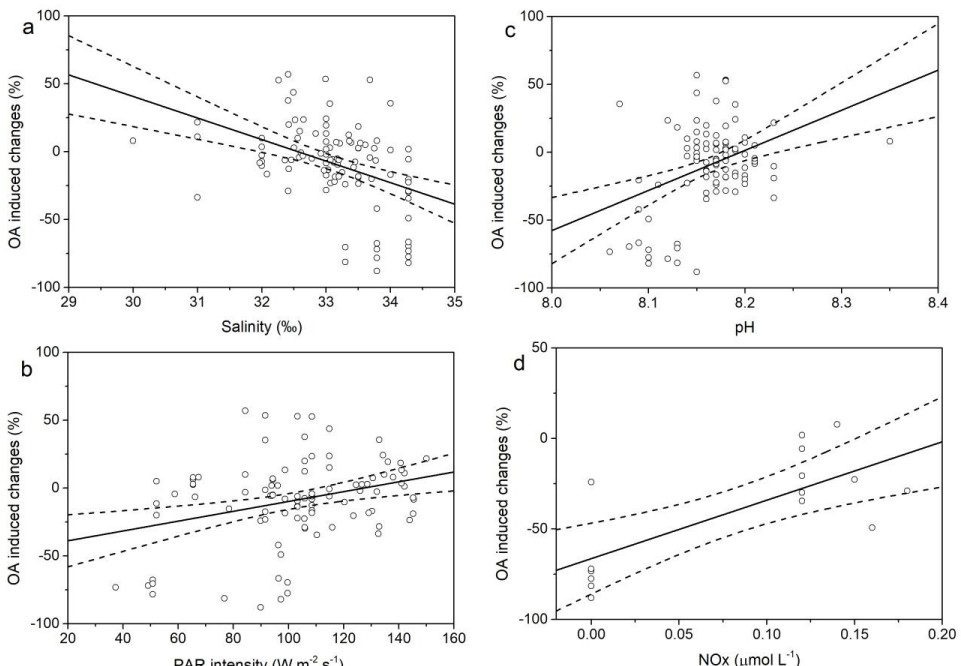

Fig. 6