# Peer review of "Contrasting responses of phytoplankton productivity between coastal and offshore surface waters in the Taiwan Strait and the South China Sea to future CO₂-induced acidification"

_Biogeosciences, 2021_

## Author Response (AR1)

**General comments**

The authors present a comprehensive set of short-term carbonate chemistry manipulations of natural phytoplankton communities from coastal and pelagic stations acquired during 3 different cruises. The manuscript it generally well written and clearly structured, and present a large dataset. I have, however, major concerns about the interpretation of the data.

Response: We appreciate these comments very much, particularly for understanding the difficulty in obtaining a large number of data through in situ investigation and deck-based incubation covering a large area.

Firstly, the data is interpreted as if these were acclimated OA responses and not short term low pH assays investigating instantaneous CO2/pH effects. These are two very different things than doing acclimated OA experiments, and show the response of the physiological machinery under current CC to short-term elevation on pCO2, so an interpolation to future OA is invalid. Also, cells are probably stressed by change in environment, especially as 100% incoming light was used in this study, which is really high. Both of these aspects need to be made clear in abstract, discussion and conclusion.

Response: We have revised the manuscript based on the two points. Please see the specific response below.

My major concern is related to your interpretation of nutrient effects. While I would agree with your general hypothesis, this dataset is not at all suited to show and conclude this: In the methods section, the analytical limits for the nitrate and nitrite measurements are given as 0.1 and 0.04 µmol L-1, respectively (L112-113), adding up to 0.14 for NOx. This value is considerable higher than most of the NOx values shown in Figure 6, potentially even higher than all except 2 or 3 values. It is not clear to me whether these are the surface values, or the ones from the bottom of the upper mixed layer the authors refer to in the results section (LL210-211). Regarding the former, this is an invalid approach. The samples you have collected and experimented with are from surface waters, so you cannot use the nutrient values from the bottom of the mixed layer as a reference. The latter have nothing to do with the physiological rates occurring in your short-term incubation assays! Also, you cannot conclude that the regional differences you observed are driven by nutrients if you have only sampled for nutrients on one of the three cruises (L115-117). Accordingly, large parts of the discussion and conclusions have to be completely re-written, and the other

environmental drivers (e.g. PAR and salinity effects, that are way more significant according to Table S1!) have to be considered more carefully.

Response: We agree with the reviewer that the nutrient values from the bottom of the mixed layer cannot be used to explain primary productivity in surface seawater. We have thus removed those data. As suggested, we have re-written the discussion and conclusions, focusing the effects of pH, PAR and salinity. Please see relevant sections for details.

Seasonality vs regionality: In L92-93 the timing of the three cruises is mentioned. While the Taiwan Strait cruise occurred in July, the other two cruises took plate in September. These differences in the seasonal timing of sampling are not at all considered in the interpretation and discussion of the results. This should be included and discussed. Specifically, to which extend could the regional differences you find be caused by the differences in timing of sampling? Also the term 'region' and the separation between nearshore vs pelagic is not clear, you need to be more specific on how these were defined.

Response: We have added the discussion of seasonality and it reads "It is worth noting that seasonality may also lead to the differential effects of SA on primary productivity since the Taiwan Strait cruise was conducted in July and the cruises of the South China Sea basin and the West South China Sea were conducted in September. The SST and solar PAR intensity of the Taiwan Strait in July was 2–3 $^{o}$C and 22 $\pm$22 W m$^{-2}$ s$^{-1}$ higher than that in September (Zhang et al., 2008, 2009; Table S3). Although the effects of SA were not related to temperature as shown in this study (Table S2), the higher solar radiation in July may contribute to the positive effect of SA on primary productivity" at line 367. We have also redefined the "region" as suggested, we now use the terms of "the continental shelf (0–200 m, 22 stations) and the slope (200–3400 m, 44 stations) and the vast deep-water basin (> 3400 m, 35 stations). In the continental shelf, the areas with depth < 50 m are defined as coastal zones (9 stations)" instead of nearshore and pelagic at line 105.

Zhang C,Zhang X, Zeng Y, Pan W, Lin J. Retrieval and validation of sea surface temperature in the Taiwan Strait using MODIS data. Acta Oceanologica Sinica, 30:153-160, 2008.

Zhang C,Ren Y, Cai Y, Zeng Y, Zhang X.Study on local monitoring model for SST in Taiwan strait based on modis data. Journal of Tropical Meteorology, 25:73-81, 2009.

Currently the carbonate chemistry data from the incubations is missing, so that it is impossible to judge if the treatments were successful. Without them, an interpretation of the results is not possible.

Response: We have added these data and the relevant description reads "A series of onboard CO2-enrich experiments in the investigated regions were conducted during three cruises. In the high $CO_2$ treatments, $pH_{total}$ had a decrease of 0.34–0.43 units, while $pCO_2$ and $CO_2$ had an increase of 676–982 µatm and 17–25 µM kg$^{-1}$ SW, respectively (Table S1). Carbonate chemistry parameters after 24 h of incubation were stable ($\triangle$pH < 0.06, $\triangle$TA < 53 µmol kg$^{-1}$ SW), indicating the successful manipulation (Table S1)" at line 215.

**Specific comments**

L1-3: include inf that this is short-term exposure and not acclimation into the title

Response: The title has been revised to "Contrasting responses of phytoplankton productivity between coastal and offshore surface waters in the Taiwan Strait and the South China Sea to short-term seawater acidification"

L25-28: As explained above, this statement is not valid

Response: The ocean acidification has been corrected to seawater acidification throughout the text.

L28-32: I don't think you can make such general statements on long-term OA effects based on 24h incubations that did not allow for any acclimation. This need to be rewriting accordingly.

Response: It has been corrected to "Contrasting responses of phytoplankton productivity in different areas suggest that SA impacts on marine primary productivity are region-dependent and regulated by local environments" at line 30.

L36-38: I suggest including that this process is ongoing and likely intensifying.

Response: Corrected.

L102ff: In Biogeosciences, total alkalinity us usually abbreviated as A_T. Please use this term throughout the manuscript.

Response: We presume that the reviewer's suggestion is to change "TAlK" to "TA". We have corrected it throughout the text.

L126-129: A lot more info needs to be given regarding the methods of OA manipulation. Was the CO2.saturated sweater taken from the same location as the sample? If yes, how much time passed between sampling and start of incubation? How was the carbonate system manipulated (e.g. TA or DIC manipulation?), the decrease of pH units by approx. 0.4 units sounds like a unprecise approach. You need to at least provide a table in the appendix with pH values at the start and the end of each incubation to prove that your OA treatments were successful, ideally a fully constrained carbonate system with measured TA and DIC values.

Response: As suggested, the text has been clarified to "Seawater that was collected from the same location as PP and filtered by cellulose acetate membrane (0.22 μm) was used to make the $CO_2$-saturated seawater, which was made by directly flushing with pure $CO_2$ until pH reached around 4.50. When saturated-$CO_2$ seawater was added to the HC treatment, equivalent filtered seawater (without flushing with $CO_2$) was also added to the AC treatment as a control. The ratios of added saturated-$CO_2$ seawater to incubation seawater were about 1:1000. Seawater was incubated within half an hour after they were collected" at line 143. We have also supplied a table of carbonate system as suggested. The relevant description reads "A series of onboard $CO_2$-enrich experiments in the investigated regions were conducted during three cruises. In the high $CO_2$ treatments, $pH_{total}$ had a decrease of 0.34–0.43 units, while $pCO_2$ and $CO_2$ had an increase of 676–982 μatm and 17–25 μM $kg^{-1}$ SW, respectively (Table S1). Carbonate chemistry parameters after 24 h of incubation were stable ($\triangle$ pH < 0.06, $\triangle$TA < 53 μmol $kg^{-1}$ SW), indicating the successful manipulation (Table S1)" at line 215.

L133: 100% incident irradiance is really high, as samples don't get mixed down in an incubator. Please consider and discuss if OA effects may be driven by high light stress in those incubations with high PAR intensity.

Response: We have added this point and it reads "It is worth noting that the samples were not mixed down in the water bath and the 100 % incident solar irradiances may have high light stress on cells. Lower incident solar irradiances or some devices can be used to simulate seawater mixing in future studies" at line 303.

L133-134: On many ships, underway seawater supply still ends up being considerable warmer that SST due to the water running through a ship. Can you provide measurements of incubator temperature and offset to SST?

Response: That is true. The following information has been supplied "Due to heating by the deck, the temperatures in the water bath were 0-2 °C higher than in situ surface seawater temperatures" at line 155.

L178: Results from nutrient measurements are missing

Response: Nutrient levels in the surface water at most stations were undetectable. That is why we used the values at the bottom of upper mixing layers. We removed the data of NOx since it is invalid to use them in explaining PP of surface seawater.

L192-193: please adjust to 'we observed that instantaneous effects of elevated pCO2'

Response: Corrected.

L194: not sure which regions were compared. Please clarify.

Response: This is a general description of OA effects in all investigated regions. It has been clarified to "It was observed that instantaneous effects of elevated $pCO_2$ on primary productivity of surface phytoplankton community in all investigated regions ranged from -88% (inhibition) to 57% (promotion), revealing significant regional differences (ANOVA, $F_{(100, 404)} = 4.103$, $p < 0.001$, Fig. 5). Among 101 stations, 70 stations showed insignificant SA effects. SA increased PP at 6 stations and reduced PP at 25 stations" at line 220.

L197: 'was' should read 'were'

Response: Corrected.

L199: should read 'approaching the Mekong River plume' and 'A reduction'

Response: Corrected.

L203-206 and elsewhere: Again this region-related effect is not 100% clear, be more precise here, please. How did you define pelagic vs. near-shore, and is this the same definition you always use when talking about regions?

Response: We have redefined the regions and it reads "Investigation areas include the continental shelf (0–200 m, 22 stations) and the slope (200–3400 m, 44 stations), and the vast deep-water basin (> 3400 m, 35 stations). In the continental shelf, the areas

with depth < 50 m are defined as coastal zones (9 stations)" at line 104. Therefore, the text here has been revised to "Overall, the elevated $pCO_2$ had neutral or positive effects on primary productivity in the continental shelf and slope regions, while having adverse effects in the deep-water basin" at line 233.

L210-211: As explained above, this NOx statistics approach is not valid

Response: We have removed the NOx data as suggested.

L219-220: Based on your data this statemen cannot be made!

Response: We have removed the information of nutrient and it reads now "Our results suggested that the enhanced effects of the SA treatment on photosynthetic carbon fixation depend on regions of different physicochemical conditions, including pH, light intensity and salinity" at line 247.

L234: should read 'reducing the removal rate'

Response: Corrected.

L237: RuBisCO abbreviation written differently in introduction. Please make consistent

Response: Corrected.

L245-276: all of this needs to be removed as you cannot conclude anything on nutrient effects based on your dataset.

Response: We have removed all of this as suggested.

L295-299: This statement cannot be made based on your dataset, rewrite following your actual data.

Response: We have rewritten it that reads "By investigating the impacts of the elevated $pCO_2$ on PP in the Taiwan Strait and the SCS, we demonstrated that such short SA-treatments induced changes in PP were mainly related to pH, light intensity and salinity based on Pearson correlation coefficients" at line 376.

Reviewer's comments for

Ms. Ref No.: bg-2021-326

Title:      Contrasting responses of phytoplankton productivity bet 1 ween coastal and offshore surface waters in the Taiwan Strait and the South China Sea to future CO2-induced acidification

Authors:      Guang Gao et al.

Submitted to: Biogeosciences

General comments

This paper examines the impacts of ocean acidification on primary productivity using a number of deck incubation experiments. A dataset of 101 experiments is unique and valuable. I consider that an approach using multiple experiments under a protocol is reasonable to assess phytoplankton responses to ocean acidification. The dataset can be useful to draw some response patterns of primary productivity to ocean acidification under any key environmental conditions. However, I found some issues in the current manuscript, so these should be clarified before publishing this paper in the journal Biogeosciences.

Response: We very much appreciate these comments and have revised the manuscript based on the reviewer's suggestions.

Major comments

1. The authors claim that higher NOx concentrations are responsible for the primary productivity enhancement under ocean acidification conditions. However, I consider that we cannot differentiate direct effect of NOx levels on phytoplankton physiology from the effect of different phytoplankton community structures derived from different NOx availabilities. Furthermore, NOx data are very few relative to other parameters observed in this study. NOx data are not at the surface but at the bottom of surface mixing layers. No direct link between the surface PP and NOx at the bottom of surface mixing layer are not explained in the manuscript. I do not consider that a conclusion is not reasonably supported by the results presented in this study.

Response: We have realized this issue and removed the data of NOx as suggested by the two reviewers.

Specific comments

2. L17 and L46. The authors mention "multiple drivers", but this study treats only one driver of ocean acidification (pH). This mention does not fit to this study's introduction.

Response: We did treat only one driver of seawater acidification but investigate its effects in different environments (temperature, salinity, solar irradiances, nutrients, etc.). To take the reviewer's suggestion, we have revised the text to "In order to examine effects of SA in changing environments" at line 17 and "it is important to understand the responses of the key players of marine biological $CO_2$ pump, the phytoplankton, to seawater acidification" at line 53.

3. L23-24. Are the digits of four reasonable?

   Response: We have changed to two digits throughout the text.

4. L95-96. It is helpful to show the number of stations examined in each category of the coastal zone, continental self, slope, and deep basin.

Response: It has been revised to "Investigation areas include the continental shelf (0–200 m, 22 stations) and the slope (200–3400 m, 44 stations), and the vast deep-water basin (> 3400 m, 35 stations). In the continental shelf, the areas with depth < 50 m are defined as coastal zones (9 stations)" at line 104.

5. We need to know how the pH electrode was treated after the calibration. Just after the calibration using NBS buffers, electrodes usually do not work appropriately in seawaters because of the quite different ion intensity between the NBS buffers and seawaters. Such electrodes may be kept in a seawater for a certain period.

   Response: What the reviewer mentioned is exactly right. We usually keep the electrode in seawater for half an hour before the measurement. This information has been added and it reads "After the calibration, the electrode of pH meter was kept in surface seawater for half an hour and then the formal measurements were conducted" at line 114.

6. How did the authors define the quantitative limits?

   Response: It was determined by compared measured values with certified reference. However, this part was removed since the reviewer 1 does not think it make sense.

How were the nutrient samples stored until analysis?

Response: The nutrient samples were measured within 24 h upon sampling and stored in a 4 $^oC$ refrigerator before analysis. However, this part was removed since the reviewer 1 does not think it make sense.

7. Was the CO2-saturated seawater filtered through 0.22 um before saturating? How was the CO2-saturated seawater made? The authors should describe the typical volume of the added CO2-saturated seawaters with the volume of pre-filtered seawater for incubations.

Response: It has been clarified to "Seawater that was collected from the same location as PP and filtered by cellulose acetate membrane (0.22 μm) was used to make the $CO_2$-saturated seawater, which was made by directly flushing with pure $CO_2$ until pH reached around 4.50. When saturated-$CO_2$ seawater was added to the HC treatment, equivalent filtered seawater (without flushing with $CO_2$) was also added to the AC treatment as a control. The ratios of added saturated-$CO_2$ seawater to incubation seawater were about 1:1000" at line 143.

8. The authors should explain the reason of selecting 0.4 pH unit.

Response: It has been clarified to "To study the impacts of projected SA (dropping by ~0.4 pH) by the end of this century (RCP8.5) on marine primary productivity in different areas (Gattuso et al., 2015)" at line 987.

9. In my understandings, whether 14C incubation for 24 h measures net production or not is under debate. In such condition, CO2 enrichment in HC treatments may alter 14C incorporation manner in phytoplankton.

Response: We would like to take this suggestion and remove the term of "net". In HC treatments, there are more $CO_2$ and $HCO_3^-$ and less $CO_3^{2-}$ compared to AC treatments, which may alter 14C incorporation manner. However, 14C exists in all carbonate forms and theoretically the ratio of 14C to 12C is consistent in all forms. Therefore, this alteration of carbonate system may not affect the accuracy of carbon fixation using 14C technique.

10. chlorophylls a --> chlorophyll a

Response: Corrected.

11. The unit for Talk of umol/L?

Response: It has been corrected to "µmol kg$^{-1}$ SW".

12. The unit for PAR of W/m2/s?

Response: The unit of W/m2/s can be used for light intensity, particularly for solar radiation (Wild et al., 2005; Yuan et al., 2021; Proutsos et al., 2022)

Proutsos, N., Alexandris, S., Liakatas, A., Nastos, P. and Tsiros, I.X., 2022. PAR and UVA composition of global solar radiation at 72a high altitude Mediterranean forest site. Atmospheric Research, 269, 106039

Wild, M., Gilgen, H., Roesch, A., Ohmura, A., Long, C.N., Dutton, E.G., Forgan, B., Kallis, A., Russak, V. and Tsvetkov, A., 2005. From dimming to brightening: Decadal changes in solar radiation at Earth's surface. Science, 308(5723), 847-850.

Yuan, M., Leirvik, T. and Wild, M., 2021. Global trends in downward surface solar radiation from spatial interpolated ground observations during 1961–2019. Journal of Climate, 34(23), 9501-9521.

13. The results of manipulation in carbonate systems should be described first. The readers need to know actual pH reduction, pCO2 and CO2 increase. The 0.4 pH reduction, even if this is identical for all experiments, would result different pCO2 increase depending on ambient temperature, salinity, and carbonate system.

Response: We have supplied relevant description and it reads "A series of onboard $CO_2$-enrich experiments in the investigated regions were conducted during three cruises. In the high $CO_2$ treatments, $pH_{total}$ had a decrease of 0.34-0.43 units, while $pCO_2$ and $CO_2$ had an increase of 676-982 µatm and 17-25 µM kg$^{-1}$ SW, respectively (Table S1). Carbonate chemistry parameters after 24 h of incubation were stable ($\triangle pH < 0.06$, $\triangle TA < 53$ µmol kg$^{-1}$ SW), indicating the successful manipulation (Table S1)" at line 215.

14. The samplings of the bottom of upper mixing layers are not described in Materials and Methods section.

Response: This part has been removed based on the comments of reviewer #1.

15. Regardless of the importance of NOx availability in this study (L220), spatial NOx distribution was not shown. Furthermore, NOx data are shown only 15, few data relative to other parameters, and this should be explained in the Materials

and Methods section. Six out of 15 data are located out of 95% confidence intervals in Fig. 6. Does this situation actually support your conclusion?

Response: We have realized this issue and removed the data of NOx.

16. The paper shows nutrient data only for NOx. We would like to know phosphate, silicic acid, ammonium, and probably iron levels.

Response: We agree with the reviewer that other nutrient elements may also affect the effects of OA on primary productivity. However, due to the limit of manpower and finance, other nutrient elements were not measured because most of them have very low levels in surface seawater of the South China Sea, requiring advanced instruments and well-trained technicians.

17. This discussion with a positive effect on diatoms is inconsistent with the discussion on a negative effect on diatoms in L237-238. Interpretations of the impact of OA on diatoms are not so straightforward as discussed here.

Response: Honestly, we are not sure which part is inconsistent with the discussion on a negative effect on diatoms in L237-238. In this study, OA showed positive, neutral and negative effects on primary productivity. We discussed the possible reasons that resulted in different effects. To clarify it, we have added some words and it reads now "Therefore, the net impact of OA depends on the balance between its positive and negative effects, leading to enhanced, inhibited or neutral influences, as reported in diatoms (Gao et al., 2012, Li et al., 2021) and phytoplankton assemblages in the Arctic and subarctic shelf seas (Hoppe et al., 2018), the North Sea (Eberlein et al., 2017) and the South China Sea (Wu and Gao 2010, Gao et al., 2012). The balance of positive and negative effects of OA could be regulated by other factors, including pH, light intensity, salinity, population structure, etc. (Gao et al., 2019a, b; Xie et al., 2022)." at line 268.

18. The "harmful algal blooms" is sudden and unacceptable, not discussed in the Discussion section.

Response: It has been removed.

19. 2. I am not sure whether the unit of total alkalinity is umol/L and that of PAR is W/m2/s.

Response: As responded above, we have changed the unit of TA to $\mu mol\ kg^{-1}$ SW and hope to keep the unit of PAR as $W/m^2/s$.

20. 5 caption. L534-536. This is described only for this figure and can be removed because of the repetition of the Result section.

Response: It has been removed as suggested.

---

## Author Response (AR2)

**Comments to the author**:
Dear Prof. Gao,

We have received two reviews of your revised manuscript from the referees who reviewed the former version of your manuscript. Both referees were pleased to find significant improvements, but they still have some concerns (see below). So we would appreciate it very much if you could make a further revised manuscript following their helpful comments.

Thank you again for your excellent efforts to improve the manuscript.

Kind regards,

Koji Suzuki
Associate Editor

Response: We appreciate the Associate Editor and two anonymous reviewers very much for their help in improving our paper. We have further revised our manuscript based on the reviewers' comments.

- Referee #1

General comments
The paper is significantly changed in response to review comments. Although the dataset shown in this paper is valuable, the authors fail to show a powerful conclusion. The paper still draws attention in this field's researchers but may not be enough for the journal Biogeosciences.

Major comments
1. Conclusions, L313. "short SA treatments induced changes in PP were mainly related to pH, light intensity and salinity" Yes, Fig. 6 shows us this. However, I consider that these are not causal relation. Phytoplankton community structures will determine SA induced changes as discussed in Discussion section. In this regard, the present conclusion may be weak.
Response: We agree with that. The following sentence has been added "In addition, phytoplankton community structures may also modulate SA induced changes".
Specific comments
2. L20. In the present abstract South China Sea is not needed to be abbreviated because the term is used only here.
Response: South China Sea and SCS were interchangeably used in the text. To be consistent, we have changed all SCS to South China Sea.
3. L60. In this manuscript abbreviations are not well managed. For example, the "SCS" appears first time here in the main text. SCS is defined firstly in line 67 and secondly

in line 72.

Response: Please see the response above.

4. L78-79. Ocean acidification --> SA?

Response: Corrected.

5. L175 and 196. uM/kg --> umol/kg

Response: Corrected.

6. L265-268. SA "increased" photosynthetic carbon fixation of three diatoms under lower light intensities but "increased" it under higher light intensities.

Response: It has been corrected to "SA increased photosynthetic carbon fixation of three diatoms (*Phaeodactylum tricornutum, Thalassiosira pseudonana* and *Skeletonema costatum*) under lower light intensities but decreased it under higher light intensities" at line 270.

7. L275-276. "100 % incident solar irradiances may have high light stress on cells", but PP was enhanced under higher PAR? Are these consistent?

Response: We added this statement as required by a reviewer. It does seem inconsistent with the results. We have revised it to "It is worth noting that the samples were not mixed down in the water bath in the present study and exposed to 100% incident solar irradiances. Lower incident solar irradiances or some devices can be used to simulate seawater mixing in future studies" at line 281.

8. L337. Some papers in the list are not referred to in the revised text.

Response: We appreciate the careful review of the referee. We have double checked the references and made corresponding corrections.

9. Fig. 2 caption. The unit for TA is still umol/L. The scale of pH should be shown.

Response: Corrected.

10. Fig. 5 and its caption. The authors mainly use the term seawater acidification rather than ocean acidification in the revised manuscript. Here also should be the case. Similar is the case for Fig. 6 and its caption. In Fig. 6 salinity should be non-dimensional.

Response: Corrected.

- Referee #2

The authors have done a very good job addressing my and the other reviewers concerns and criticism, so that the manuscript improved a lot in the revised version. There are just a few minor edits I would suggest.

Response: We appreciate this comment and have further revised the manuscript as suggested.

L28: should read 'vulnerable to a drop'

Response: Corrected.

L57: The differences of SA relative to the before mentioned OA needs to be explained/made explicit, it is currently not clear that the authors use SA for short-term changes in carbonate chemistry

Response: The text has been clarified to "In addition to the slow change of ocean acidification, some processes, such as freshwater inputs, upwelling, typhoon and

eddies, can lead to instantaneous $CO_2$ rising and short-term changes in carbonate chemistry, termed seawater acidification (SA) (Moreau et al., 2017; Yu et al., 2020)" at line 41.

L78-83: Jumping back and forth between OA and SA, this needs to be made consistent, or explicit (i.e. if referring to different time scales)

Response: Corrected.

L126: should read 'reached values around 4.50'

Response: Corrected.

L127-130: the abbreviations AC and HC need to be explained and written out when they first appear

Response: Corrected.

L129: should read 'Samples were incubated'

Response: Corrected.

L188: should read 'productivity ranged from 99 to 302'

Response: Corrected.

L189: should read 'from 17 to 306'

Response: Corrected.

L195: please always use the same names for your treatment throughout the manuscript

Response: Corrected.

L195-196: should read 'pH_total decreased by […], while pCO2 and O2 increased by'

Response: Corrected.

L199-202: similar to the first version of this manuscript, it is still not clear which regions have been compared for the statistical test. Please clarify that you compare continental shelf, slope and deep-water basin stations here

Response: It has been clarified to "It was observed that instantaneous effects of elevated $pCO_2$ on primary productivity of surface phytoplankton community in all investigated regions ranged from -88% (inhibition) to 57% (promotion), revealing significant regional differences among continental shelf, slope and deep-water basin (ANOVA, $F_{(2, 98)} = 3.747$, $p = 0.027$, Fig. 5)" at line 202.

L208: space missing in 'to14'

Response: Corrected.

L214-217: In a linear regression analysis, the p value only indicated whether the slope of the fit is different from 0, but does not give any indication on how good the fit is. R values of 0.3-0.4 are low, and the fact that most of the data points in figure 6 lay outside of the confidence intervals nicely illustrates this. Therefore, I would not trust these SA-effects too much. See later comment.

Response: We agree with this.

L253: 'limited' should read 'limiting'

Response: Corrected.

L261-262: should read 'The nutrient levels in the basin are usually lower than on the shelf'

Response: Corrected.

L264: should read 'with increasing light intensity'

Response: Corrected.

L264-268: Given the weak correlation (r=0.31) I would not overinterpret this observation

Response: We agree with that. This point has been underlined and the text reads " Meanwhile, the weak correlation ($r = 0.311$) between light intensity and SA effect suggests the deviation from linear relationship in the context of multiple variables needs to be further illuminated in future studies" at line 279.

L277: should read 'A negative correlation'

Response: Corrected.

L283-286: I don't fully understand this sentence, this needs to be formulated more clearly. Are you trying to say that the correlation with salinity seems to be an autocorrelation between salinity and insitu pH? If yes, please say so explicitly. This sentence also needs some grammar editing.

Response: The text has been clarified to "In this study, the negative relationship between salinity and SA effects seems to be an autocorrelation between salinity and in situ pH (Fig. S1) because lower salinity occurred in coastal waters where seawater pH was higher while the basin zone usually had higher salinities and lower pH" at line 291.

L293-294: should read 'with high abundances of phytoplankton, which is consistent'

Response: Corrected.

L303-306: it is not clear what you mean with the term 'seasonality', as you are now discussing some of the environmental variables in more details before. I think species succession should be explicitly mentioned

Response: One reviewer asked us to discuss the effect of seasonality, because the Taiwan Strait cruise was conducted in July while the cruises of the South China Sea basin and the West South China Sea were conducted in September, which may be classified as summer and autumn, respectively. As suggested, we have also discussed the species succession and it reads "In addition, species succession of phytoplankton with season may also affect the response to SA (Xiao et al., 2018)" at line 320.

L318-320: it needs to be made explicitly mentioned that your predictions only hold true if responses to very short-term pH changes are representative for responses to long-term OA trends. I also find these sentences a bit to broad, I don't think you can claim based on your data that 'PP in coastal waters would be increased', there are e.g. a lot of unaffected stations. These statements need to be toned down significantly.

Response: We agree with the reviewer. These statements have been toned down to "The negative effect of SA in basin zones may further reduce primary productivity. Meanwhile, PP in some coastal waters may be increased by SA" at line 330.

---

## Author Response (AR3)

**Comments to the author**:
We are pleased to inform you that your manuscript has been accepted for publication in Biogeosciences. Thank you very much for your tremendous efforts in revising the manuscript. Before publishing this paper, please consider my technical suggestions below.

Thank you again for choosing the journal Biogeosciences.

Kind regards,

Koji Suzuki
Associate Editor

Response: We appreciate the Associate Editor, Prof. Koji Suzuki, very much for his kind help in improving our paper. We have corrected our manuscript based on the suggestions below.

L16: the effects of SA
L20: the deep-water basin
L42: typhoons
L45: of the ocean
L57: the neutral
L66: the Taiwan Island -> the Island of Taiwan
L70: due to an increased supply
L110: was -> were
L119: collected with a 10 L
L136: the changes in carbonate systems
L177–178: the highest value in the South China Sea near the islands in the Philippines
L179–180, L311, and Fig. 2J: The unit of mean PAR intensity should be W m-2, not W m-2 s-1.
L229: responsible for the contrasts of the short-term
L258: in the deep-water basin
L260 and L261: the photosynthesis
L267: This was inconsistent with the study of Gao et al. (2012)
L272: picophytoplankton
L277: from the linear relationship
L282: changes in PP
L282: The decrease in salinity
L283: on the photosynthetic carbon fixation of the coccolithophore
L286: the effective quantum yield of microplankton community
L293: picophytoplankton
L294–295: nutrient uptake
L315: seasonal phytoplankton species succession

L322: changes in phytoplankton community structure may also modulate the SA-induced variability.

L324–325: transport of nutrients

L329: designed the research